# Longitudinal Measurement Invariance of the Dual School Climate and School Identification Scale (SCASIM-St15) in Chilean Adolescents

**DOI:** 10.3390/bs15060750

**Published:** 2025-05-30

**Authors:** José Luis Gálvez-Nieto, Ítalo Trizano-Hermosilla, Karina Polanco-Levicán, Ignacio Norambuena-Paredes, Maura Klenner-Loebel, Sandra Riquelme-Sandoval

**Affiliations:** 1Departamento de Trabajo Social, Universidad de La Frontera, Temuco 4780000, Chile; jose.galvez@ufrontera.cl (J.L.G.-N.); ignacio.norambuena@ufrontera.cl (I.N.-P.); sandra.riquelme@ufrontera.cl (S.R.-S.); 2Departamento de Psicología, Universidad de La Frontera, Temuco 4780000, Chile; italo.trizano@ufrontera.cl; 3Facultad de Educación, Universidad Autónoma de Chile, Temuco 4810101, Chile; 4Departamento de Lenguas, Literatura y Comunicación, Facultad de Educación, Ciencias Sociales y Humanidades, Universidad de La Frontera, Temuco 4780000, Chile; maura.klenner@ufrontera.cl

**Keywords:** school climate, adolescence, academic expectations, longitudinal design

## Abstract

School climate is a highly relevant construct in the educational field; however, most research adopts cross-sectional designs, which limits the understanding of its stability and development over time. Consequently, this study aimed to assess the degree of the longitudinal invariance of the SCASIM-St15 in a sample of adolescent students throughout their secondary education trajectory. A longitudinal panel design was used, with one-year intervals, covering the entire secondary education cycle and following the same cohort of 679 Chilean students across four measurements: wave 1 = 1st year of secondary school with a mean age of 14.53 Sd = 0.625; wave 2 = 2nd year with a mean age of 15.60 [Sd = 0.629]; wave 3 = 3rd year with a mean age of 16.55 [Sd = 0.602]; and wave 4 = 4th year with a mean age of 17.49 [Sd = 0.587]. The results from factorial invariance modeling across the four time points indicate that the SCASIM-St15 shows an overall good fit, with satisfactory goodness-of-fit indices, suggesting that the factorial structure of the SCASIM-St15 remains stable over time.

## 1. Introduction

School climate has garnered growing interest ([17]; [40]), considering its relevance and its connection with various factors that are key to the positive development of adolescents ([48]; [57]; [67]). School climate influences the educational process, as it results from the interaction among different members of the school community. This dynamic gives rise to affective relationships, conflicts, expectations, and beliefs that impact students’ development over time ([7]; [53]). Due to its dynamic nature, school climate can benefit students by predicting better mental health during adolescence and, consequently, greater well-being ([30]). This is especially important since adolescence is a stage in which psychological difficulties may emerge, potentially linked to the various physical, emotional, and cognitive changes that interact with individual vulnerability ([39]). In this regard, it may be essential to have an appropriate assessment of school climate, meaning it should be measured with a valid and reliable instrument that supports sound decision making and enables the implementation of appropriate interventions.

School climate is understood as the affective and cognitive perceptions held by students, teachers, school leaders, and staff regarding social interactions, interpersonal relationships, values, and beliefs within an educational institution ([51]). In this regard, school climate has been associated with better academic performance, school engagement ([30]; [58]), and a stronger sense of school belonging ([70]). Additionally, a positive school climate contributes to lower levels of school absenteeism and bullying victimization ([33]). Moreover, school climate plays a crucial role, as a positive teacher–student relationship helps reduce adolescent burnout ([28]; [66]), and it is also linked to teachers’ receptiveness to social and cultural diversity ([47]).

According to the proposal by [38] ([38]), school climate is conceived as the interaction that occurs among the various members of the educational community, such as students, teachers, and administrative staff. This perspective includes emotional connection, support, motivation, and academic expectations that emerge during the educational process, allowing students to improve their adaptation, become more engaged, and enhance their academic performance. The perception of sharing beliefs and values with the school community strengthens students’ sense of belonging and connection, leading them to identify with their educational community and school. This approach is built upon and understood through Bronfenbrenner’s Ecological Systems Theory ([7]), which posits that human development occurs within an environmental context composed of interrelated systems (microsystem, mesosystem, exosystem, macrosystem, and chronosystem). These systems, ranging from the most immediate to the most distant, influence an individual’s development.

School climate is also related to students’ personal skills, specifically to socio-emotional competencies ([48]; [57]), adolescents’ psychological adjustment ([17]), resilience ([41]), and prosocial behavior ([67]). In this way, students achieve a better quality of life ([73]), greater well-being, and improved mental health ([4]; [72]). On the contrary, a negative school climate is associated with the problematic use of smartphones and social media ([6]), along with a higher risk of depressive symptoms and experiences of discrimination against LGBTQ+ adolescents by peers and teachers ([13]; [2]), as well as against culturally distinct students ([68]). In this regard, it has been observed that secondary students with emotional and behavioral disorders have a more negative perception of school climate compared to their peers ([18]), as do students with disabilities who receive special education services ([31]; [68]). Thus, a negative school climate fosters conditions that contribute to discrimination and victimization ([16]; [29]; [42]). Additionally, school climate and school identification have a direct and significant influence on smoking or vaping behavior through the mediation of perceived stress coping ([72]). Hence, school climate plays a key role in students’ development, both in terms of prevention and in guiding interventions within educational institutions.

School climate is related to academic expectations ([1]; [14]; [24]; [46]). Academic expectations are relevant in the teaching and learning process and contribute to improved academic performance ([3]; [9]). It is important to note that positive student–teacher relationships foster adolescents’ academic expectations ([69]). In this regard, high academic expectations and a positive school climate help reduce dropout rates, as students are more receptive to their teachers when they feel valued and supported ([32]). Additionally, students who perceive high academic expectations and levels of support from their teachers tend to be more engaged with their school ([54]). Furthermore, high academic expectations are associated with a decrease in teasing and bullying at school ([36]). According to [22] ([22]), the academic expectations adolescents perceive their teachers to have constitute an individual factor that helps explain school climate.

Consequently, this theoretical and empirical proposal considers both school climate and school identification ([38]), as the bond and sense of belonging a student develops toward their school are essential, influencing their perception of the school climate as adequate or satisfactory. It is worth mentioning that social identity refers to the sense of belonging a person feels toward certain groups, involving an emotional bond and a positive evaluation of being part of the group ([55]; [62]). People identify with different groups to varying degrees, and the stronger the identification, the greater the adoption of the group’s beliefs and the acceptance and expression of its behaviors ([8]; [55]; [62]). In this regard, [38] ([38]) propose that students’ social identity is their psychological bond with their school, which tends to become a meaningful social group. It should be noted that school climate and school identification contribute to students’ academic achievement ([43]; [49]; [59]). In this regard, social identity and school climate influence each other, especially during adolescence ([37]). From perspectives such as ecological theory ([7]) and social identity theory ([55]), it is understood that a positive climate strengthens one’s sense of belonging and identity, which in turn enhances their perception of the climate, promoting their well-being, academic engagement, and life satisfaction ([5]; [45]; [56]).

[38] ([38]) operationalized their proposal through the dual-scale School Climate and School Identification Measure—Student (SCASIM-St). This instrument was developed using a sample of Australian students from the 7th to 10th grades. It presents a multidimensional second-order factorial structure labeled school climate, which consists of four first-order factors: Student–Student Relations (social relationships among peers), Student–Staff Relations (relationships between students and school staff), Academic Emphasis (support provided by teachers to help students achieve learning goals), and Shared Values and Approaches (the extent to which students share the school’s goals and rules). In addition, a fifth first-order factor, School Identification, is included, which assesses the extent to which belonging to the school is integrated into the student’s identity.

This instrument has shown evidence of validity and reliability in different countries ([15]; [20]; [38]) and is available in two versions: a full version with 38 items and a short version with 15 items ([21]; [71]). Both versions demonstrate adequate psychometric indicators. Specifically, this instrument has shown evidence of invariance in various studies. [38] ([38]) reported that the factorial structure remains consistent despite differences in gender, grade level, and linguistic background. Similarly, measurement invariance—including configural, metric, and scalar forms—was confirmed based on gender and school type among Chinese adolescents ([71]). In the case of Chilean students, the factorial structure of the SCASIM-St15 was invariant by gender up to the metric level, and invariance was maintained up to the scalar level according to type of education and age ([21]).

### Longitudinal Research on School Climate

Considering the relevance of this construct and the observations of [26] ([26]), who notes that a lack of longitudinal studies limits school climate research, it is important to consider the findings of the investigations that assess school climate over time, as they allow for the establishment of causal relationships between school climate and other constructs. [64] ([64]) analyzed the role of school climate and the changes that occur in motivation and academic achievement, indicating that students’ positive perceptions of school climate at the beginning of the study (Time 1) were positively associated with changes in self-determined academic motivation at the midpoint (Time 2), which in turn were positively related to changes in grades at the end of the study (Time 3), beyond the effects of gender and age. Similarly, other studies have reported that a negative school climate is associated with lower academic achievement in adolescents ([44]; [50]). [26] ([26]) also conducted a two-year longitudinal study, indicating that perceptions of a better school climate predicted greater emotional engagement and lower levels of burnout after one year. Additionally, school climate was found to mediate the relationship between school-based racial socialization and suspensions among black and white students in a three-year study ([65]).

Along the same lines, school climate is important for facing crises and unexpected changes. In this regard, a four-year Australian longitudinal study (2018–2021) showed that during the COVID-19 pandemic, student engagement in learning activities and well-being declined significantly. However, schools with a positive school climate and in which students identified with the school prior to the pandemic showed better academic performance ([11]). [52] ([52]) conducted a three-year longitudinal study (2017–2019) with students in grades 7–8 and concluded that a positive school climate predicts greater school identification and resilience in adolescents. Similarly, [35] ([35]) found that school identification mediates the association between school climate and student well-being, while [70] ([70]) provided additional evidence that a negative school climate is associated with a higher risk of depressive symptoms in adolescents.

Longitudinal studies have also shown a significant link between school climate and students’ mental health. In this regard, a two-year study that assessed adolescents every six months reported that changes in perceptions of school climate preceded changes in gaming disorder symptoms among preadolescents. On the other hand, students who do not feel satisfied with the school they attend show a greater tendency to use substances, both cross-sectionally and longitudinally. Therefore, it is concluded that a positive school climate could protect adolescents from substance use ([50]). According to [70] ([70]), a negative school climate at Time 1 was significantly and negatively associated with the perception of belonging to the school, while at Time 2 it was positively associated with depressive symptoms. Similarly, greater school support in eighth grade had direct impacts, reducing depressive symptoms and increasing academic competence and self-efficacy by 10th grade. Consequently, a positive school climate helps prevent demotivation, which benefits academic performance and adolescent mental health ([27]).

School climate is essential for supporting students’ mental health, well-being, and learning, enhancing their overall development. Schools are considered ideal settings for implementing prevention and promotion strategies to foster a positive school climate as part of broader efforts to educate adolescents and train teachers ([4]). To this end, it is crucial to conduct an initial assessment to understand students’ perceptions of the school climate in their institution, so that relevant planning can be developed and actions can be implemented that truly make a meaningful contribution. Consequently, it is important to highlight that school climate should be assessed using instruments with sound psychometric properties. This would enable more accurate baseline and follow-up evaluations, and therefore, better-planned, more relevant, and more successful intervention programs that benefit both students and the educational community.

The relevance of this research lies in the need to assess whether the instrument measures the same construct over time, which allows us to affirm that the changes observed in students’ scores across successive applications reflect variations in participants’ perceptions of school climate, rather than being better explained by differences in the interpretation of the scale’s items. Additionally, it enables trust in the stability of the results, thus making it possible to evaluate the impact of programs that include pre- and post-intervention assessments, in which it is essential that the scores obtained by adolescents on school climate are not better explained by measurement errors. It also allows for the analysis of scores across different groups, making it possible to carry out comparisons and to plan and implement appropriately tailored and relevant actions.

Based on the aforementioned points, this study aimed to assess the degree of longitudinal invariance of the SCASIM-St15 in a sample of adolescent students throughout their secondary education trajectory. Based on this objective, the following hypotheses were proposed:

**H1:** *The SCASIM-St15 will exhibit a five-factor correlated factorial structure that remains stable in each of the four cross-sectional waves of measurement*.

**H2:** *The SCASIM-St15 will demonstrate configural factorial invariance over time, indicating that the five-factor correlated model remains equivalent across the four measurement waves*.

**H3:** *The SCASIM-St15 will demonstrate metric invariance over time, indicating that the factor loadings remain equivalent across the four measurement waves*.

**H4:** *The SCASIM-St15 will demonstrate scalar invariance over time, indicating that the item intercepts remain equivalent across the four measurement waves*.

**H5:** *The internal consistency reliability of the SCASIM-St15 will be adequate across the four measurement waves*.

## 2. Materials and Methods

### 2.1. Participants

The study included an initial cohort of 679 Chilean adolescent students from 26 educational institutions distributed across five regions of the country, representing all the main geographical areas of Chile. These students were followed across four waves of data collection, starting in their first year of secondary education in 2019 and concluding in their fourth year in 2023. The design corresponded to a longitudinal panel, which means that the same group of students was assessed at each measurement point, allowing for the analysis of changes over time within the same individuals.

The sample consisted of students of both sexes (38.1% male and 61.9% female), with a mean age of 14.53 years (SD = 0.625) in the first measurement. Additionally, 19.8% of participants identified with an ethnic group present in Chile.

Although minor attrition was recorded throughout the study (less than 10%), the final sample analyzed remained sufficiently stable, and no systematic biases attributable to dropout were identified. Therefore, the changes observed over time can be confidently attributed to developmental or contextual factors rather than to variations in sample composition.

### 2.2. Instruments

Three instruments were administered to achieve the objective of this study. The first instrument was a sociodemographic questionnaire consisting of a set of closed-ended questions, for example: sex, age, nationality, ethnic background, family origin (urban/rural), name of the educational institution, and school grade, among other variables.

In addition, the Chilean short version of the Climate and School Identification Measure—Student (SCASIM-St15) was administered ([21]). The SCASIM-St15 is a self-report scale answered using a five-point Likert scale (1 = strongly disagree, 5 = strongly agree). It includes five factors: Student–Student Relations, Student–Staff Relations, Academic Emphasis, Shared Values and Approaches, and School Identification. The SCASIM-St15 demonstrated adequate levels of validity based on CFA goodness-of-fit indices: WLSMV-χ^2^ (660) = 3528.580, *p* < 0.001; CFI = 0.964; TLI = 0.961; and RMSEA = 0.049 (90% CI = 0.048–0.051). Reliability indices were also satisfactory, with McDonald’s omega values ranging from ω = 0.914 for the “School Identification” factor to ω = 0.856 for the “Shared Values and Approaches” factor.

### 2.3. Procedure

To implement the research protocols, school principals were contacted and asked to sign a collaboration agreement to gain access to the sample. Once authorization was obtained, informed consent forms were sent to the students’ parents or guardians, and subsequently, students provided informed assent. Finally, the questionnaires were administered voluntarily, preferably during the morning hours, ensuring adherence to the ethical principles approved by the Scientific Ethics Committee of the University of La Frontera, Chile.

### 2.4. Data Analysis

Descriptive analyses were conducted on the 15 items of the instrument used in this study, obtaining statistics for mean, standard deviation, skewness, and kurtosis in each of the four measurement waves. Then, the internal structure was evaluated cross-sectionally across the four time points. For this purpose, the MLR estimation method was used ([34]). Model quality was assessed using the following fit indices: Comparative Fit Index (CFI), Tucker–Lewis Index (TLI), Root Mean Square Error of Approximation (RMSEA), and Standardized Root Mean Square Residual (SRMR). For the first two indices, the model was considered to adequately reproduce the data when values exceeded 0.95. In contrast, for the latter two, the model was considered excellent when values were below 0.06 and 0.08, respectively.

Once the internal structure of the instrument was confirmed, longitudinal factorial invariance was progressively evaluated through three models: configural (baseline model), metric (factor loading invariance), and scalar (intercept invariance). To test invariance, each model with additional constraints was compared to the baseline model using the DIFFTEST procedure. In addition to the chi-square difference test, changes in the fit indices of CFI, TLI, and RMSEA were examined to assess model equivalence. Following the recommendations of [12] ([12]), a change (Δ) in CFI less than or equal to 0.01 was considered evidence of invariance. Additionally, in line with the criteria proposed by [10] ([10]), a change in TLI less than or equal to 0.01 and a change in RMSEA below 0.015 were also considered indicative that the restricted model did not significantly worsen the fit, thus supporting the invariance hypothesis.

Finally, reliability estimates for the instrument scores were obtained using the alpha, omega, and GLB coefficients cross-sectionally for each of the four waves of measurement ([60]). Descriptive and reliability analyses were conducted using JASP version 0.18.3, while factorial and invariance analyses were carried out using Mplus version 8.11.

## 3. Results

### 3.1. Descriptive Statistics

Table 1 presents the descriptive statistics of the items evaluated across four measurements, including means, standard deviations, skewness, and kurtosis. Overall, a progressive increase in the means of most items can be observed from Time 1 to Time 3, followed by a decrease at Time 4. For example, Item 8 shows the highest means across all measurements, reaching its peak at Time 3 (M = 4.31; Sd = 0.73). This item displays negative skewness (−1.37) and positive kurtosis (3.70), indicating a distribution skewed toward the higher values of the scale and a greater concentration around the mean.

### 3.2. Longitudinal Invariance

The longitudinal invariance analysis began with the estimation of a confirmatory factor model at each of the four cross-sectional time points. As shown in Table 2, the goodness-of-fit indices demonstrated satisfactory values at all time points (RMSEA ≤ 0.044, CFI ≥ 0.975, TLI ≥ 0.958, SRMR ≤ 0.033), indicating the stable factorial structure of the SCASIM-St15.

Regarding the results of the longitudinal invariance analysis of the SCASIM-St15 (Table 2), the configural model showed a satisfactory fit (RMSEA = 0.030, CFI = 0.957, TLI = 0.950, SRMR = 0.031), indicating that the five-factor correlated structure is consistent over time. Subsequently, the metric invariance model was analyzed, which imposes constraints on factor loadings across time points. When comparing the tested invariance models using the DIFFTEST, a non-significant difference was observed between the metric and configural models (chi-square = 37.886, df = 30, *p* = 0.1527). Therefore, metric invariance is accepted across the four measurement waves, indicating that the factor loadings are equivalent over time. Finally, the scalar invariance model was evaluated, which imposes constraints on the intercepts across the four measurements. To determine whether this level of invariance was supported, the scalar model was compared to the metric model. Although the chi-square difference was statistically significant (chi-square = 125.274, df = 30, *p* < 0.01), this criterion alone is insufficient to reject the scalar invariance model. Thus, the changes in RMSEA (0.002), CFI (−0.005), and TLI (−0.005) were considered, and these remained within acceptable thresholds according to established criteria for evaluating invariance models. This indicates that the intercepts are equivalent across the four measurement waves, supporting the scalar invariance of the SCASIM-St15.

### 3.3. Evidence of Reliability

Table 3 presents the reliability coefficients, showing adequate levels of internal consistency over time. Overall, a progressive improvement in reliability can be observed as time advances, reaching values between 0.843 and 0.916 in the fourth wave of measurement (T4). Specifically, Factor Five shows the highest values across all measurements, whereas Factor One reports the lowest values in the initial measurements. However, an increase in its coefficients is also observed in the last two measurements.

## 4. Discussion

The present study aimed to assess the degree of longitudinal invariance of the SCASIM-St15 in a sample of adolescent students throughout their secondary education. The results fully achieved this objective, providing evidence of the psychometric quality of the scale over the four years of secondary schooling.

First, cross-sectional CFAs were conducted for the four time points, followed by the evaluation of measurement invariance using a longitudinal design. In addition, satisfactory reliability evidence was presented across all periods. An interesting result is the progressive increase in the reliability coefficients of the scale across the four measurements. This increase can be partly attributed to students’ growing familiarity with the content and structure of the items, which supports more consistent responses over time. Likewise, considering the panel design and the four-year follow-up, this pattern could also reflect cognitive, emotional, and social maturation processes characteristic of late adolescence. It is worth noting that this pattern has not been reported in previous studies of the SCASIM-St, which have been predominantly cross-sectional (e.g., [23]; [38]; [15]). These results are particularly relevant for applied settings, in which diagnostic studies require high precision in measuring the latent trait.

Regarding the factorial structure of the SCASIM-St15, although previous studies on this scale have only reported cross-sectional analyses, the findings of the present study are consistent with the extended version of 38 items and five factors ([15]; [20]; [38]), as well as the reduced version with 15 items and five correlated factors ([21]; [71]). These results support the stability of the five-factor structure throughout the students’ secondary education trajectory, reinforcing the scale’s validity for use in longitudinal studies within this context.

Another relevant finding of this study is the evidence of factorial invariance for the SCASIM-St15. Our results support the psychometric quality of the scale through a longitudinal design, reaching the level of scalar invariance. These results are consistent with those reported by [25] ([25]), who also found scalar invariance in a multidimensional measure of school climate over three years in a sample of Canadian adolescents ([25]). They also expand on previous evidence regarding the factorial stability of the scale, which was previously assessed using cross-sectional designs based on gender, school grade, and linguistic background ([38]). Similarly, research on Chinese adolescents ([71]) has explored invariance concerning sex and school type, while in the Chilean context, studies have analyzed the degree of invariance according to sex, type of education, and age ([21]). These findings help fill a gap in the literature by addressing the scarcity of longitudinal studies on school climate ([26]).

The SCASIM-St15 scale significantly contributes to the study of educational environment by offering an integrated theoretical framework that connects school climate and identification ([38]). This integration highlights the importance of students’ sense of connection and belonging to their school ([55]; [62]), which positively influences the creation of a favorable school environment by promoting variables such as academic achievement ([43]; [49]; [59]) and reducing teasing and bullying at the school level ([36]). In this regard, I. [61] ([61]) demonstrated that school identification mediates the relationship between a positive school climate and a reduction in bullying. These results support the idea that fostering a strong sense of identification with the school community not only enhances engagement but also serves as a protective factor against peer victimization over time.

Future lines of research could explore the relationship between school climate and school identification with other variables relevant to adolescent development, such as psychological distress ([19]), school dropout ([63]), or school engagement ([30]; [58]). Longitudinal studies that integrate these variables would allow for the development of more comprehensive predictive models.

Regarding the limitations of this study, first, although a panel design with annual follow-up of the same cohort was used, the sample was focused on students from Chile; therefore, generalizations to other cultural or educational contexts should be made with caution. Additionally, potential contextual changes in the schools, such as curricular reforms or external events, which may have influenced the perception of school climate, were not considered. Finally, the analysis focused on structural invariance without incorporating predictive relationships with external variables, which could be addressed in future research.

Despite these limitations, the results offer relevant implications for school interventions. The evidence of longitudinal invariance suggests that the SCASIM-St15 can be reliably used to assess changes in school climate over time, which is essential for the proper implementation of institutional programs, educational improvement projects, or interventions aimed at promoting positive school climates. Having a measurement instrument that is reliable and valid over time makes it possible to identify specific variables or factors that require attention more accurately and to design targeted strategies according to the educational level or the profile of the institution. Furthermore, the progressive improvement in internal consistency observed over time may reflect a greater degree of ownership and understanding on the part of the students, which reinforces the usefulness of applying the scale as a tool to monitor school climate trajectories in educational settings.

## 5. Conclusions

This study analyzed the validity and reliability of the SCASIM-St scale throughout the school trajectory of adolescent students in Chile. Our results support the factorial structure of 15 items and five correlated factors, which are consistent with the original study and subsequent validations.

Longitudinal invariance analyses confirmed the structural stability of the scale over time, reaching the level of scalar invariance. This result supports the validity of comparisons across different points in the academic trajectory, as both factor loadings and intercepts remained consistent.

Regarding reliability, a progressive improvement was observed in internal consistency coefficients, with high values in the final measurement. This increase strengthens the robustness of the instrument for use in longitudinal studies.

Taken together, the findings support the psychometric quality of the SCASIM-St15 and its relevance for research on school climate and identification. Its application allows for monitoring the school environment and guiding interventions aimed at strengthening students’ sense of belonging and cohesion within educational institutions.

Finally, it is suggested that future research explore individual trajectories using latent growth models and examine the relationship between school climate and identification with key variables of adolescent development, such as psychological distress, dropout, and academic engagement, moving toward more comprehensive explanatory models of student well-being.

## Figures and Tables

**Table 1 behavsci-15-00750-t001:** Descriptive statistics.

	**Time 1**				**Time 2**			
**Item**	**M**	**Sd**	**g1**	**g2**	**M**	**Sd**	**g1**	**g2**
1	3.11	0.902	−0.222	−0.111	3.52	0.851	−0.354	0.342
2	3.23	0.846	−0.254	−0.007	3.46	0.781	−0.134	0.436
3	3.37	0.83	−0.223	0.110	3.58	0.799	−0.255	0.417
4	4.08	0.788	−0.777	1.035	4.21	0.732	−1.023	2.246
5	3.91	0.889	−0.776	0.703	4.00	0.808	−0.726	0.916
6	3.90	0.837	−0.756	0.932	4.01	0.818	−0.822	1.150
7	4.11	0.750	−0.562	0.049	4.19	0.768	−1.029	1.733
8	4.40	0.743	−1.348	2.288	4.42	0.706	−1.525	3.966
9	4.19	0.870	−1.008	0.855	4.22	0.824	−1.137	1.704
10	3.33	0.941	−0.219	−0.182	3.48	0.932	−0.202	−0.185
11	3.57	0.928	−0.385	0.126	3.70	0.971	−0.616	0.167
12	3.47	0.917	−0.298	−0.058	3.58	0.930	−0.347	−0.161
13	3.80	1.043	−0.698	0.001	3.93	0.971	−0.912	0.773
14	3.35	1.068	−0.255	−0.404	3.56	1.049	−0.423	−0.239
15	3.39	1.099	−0.318	−0.459	3.50	1.054	−0.385	−0.279
	**Time 3**				**Time 4**			
**Item**	**M**	**Sd**	**g1**	**g2**	**M**	**Sd**	**g1**	**g2**
1	3.75	0.797	−0.609	0.907	3.43	0.891	−0.535	0.495
2	3.59	0.794	−0.367	0.522	3.38	0.904	−0.539	0.337
3	3.65	0.812	−0.411	0.39	3.51	0.867	−0.543	0.426
4	4.15	0.763	−0.847	1.379	3.93	0.827	−0.758	1.029
5	4.02	0.803	−0.800	1.224	3.79	0.865	−0.700	0.697
6	4.04	0.800	−0.792	1.028	3.78	0.863	−0.604	0.436
7	4.13	0.788	−1.043	1.982	3.95	0.838	−0.805	1.028
8	4.31	0.731	−1.374	3.700	4.12	0.794	−1.093	2.292
9	4.18	0.842	−1.039	1.306	3.91	0.920	−0.889	0.967
10	3.60	0.918	−0.532	0.396	3.34	0.959	−0.300	−0.091
11	3.71	0.905	−0.577	0.444	3.47	0.987	−0.416	−0.017
12	3.70	0.86	−0.561	0.408	3.44	0.96	−0.365	0.046
13	3.92	0.937	−0.73	0.392	3.62	1.005	−0.598	0.237
14	3.55	1.033	−0.356	−0.235	3.32	1.091	−0.289	−0.353
15	3.50	1.061	−0.415	−0.263	3.32	1.110	−0.356	−0.336

Note: M = Mean; Sd = Standard deviation; g1 = Skewness; g2 = Kurtosis.

**Table 2 behavsci-15-00750-t002:** Longitudinal measurement invariance of the SCASIM-St15.

Model	MLR χ^2^ (df)	RMSEA	CFI	TLI	SRMR	ΔRMSEA	ΔCFI	ΔTLI	ΔSRMR	DECISION
Time 1	152.525 (80)	0.039	0.978	0.971	0.033	—	—	—	—	—
Time 2	140.804 (80)	0.036	0.986	0.981	0.027	—	—	—	—	—
Time 3	125.371 (80)	0.031	0.988	0.958	0.028	—	—	—	—	—
Time 4	171.799 (80)	0.044	0.975	0.967	0.033	—	—	—	—	—
Configural	2353.831 (1520)	0.030	0.957	0.950	0.031	—	—	—	—	Accepted
Metric invariance	2391.939 (1550)	0.030	0.957	0.951	0.033	0	0	0.001	0.002	Accepted
Scalar invariance	2524.424 (1580)	0.032	0.952	0.946	0.034	0.002	−0.005	−0.005	0.001	Accepted

Note: MLR − χ^2^ = Maximum Likelihood Robust; df = degrees of freedom; RMSEA = Root Mean Square Error of Approximation; CFI = Comparative Fit Index; TLI = Tucker–Lewis fit index; SRMR = Standardized Root Mean Square Residual; ΔRMSEA = Change in RMSEA; ΔCFI = Change in CFI; ΔTLI = Change in TLI; ΔSMRM = Change in SMRM.

**Table 3 behavsci-15-00750-t003:** SCASIM-St15 reliability across the four measurement times.

Wave	Reliability Coefficient	F1	F2	F3	F4	F5
Time 1	Omega	0.740	0.814	0.784	0.758	0.908
	Alpha	0.739	0.804	0.775	0.754	0.906
	GLB	0.740	0.814	0.784	0.758	0.908
Time 2	Omega	0.790	0.845	0.812	0.833	0.921
	Alpha	0.789	0.842	0.812	0.833	0.919
	GLB	0.790	0.845	0.812	0.833	0.921
Time 3	Omega	0.832	0.872	0.829	0.836	0.916
	Alpha	0.832	0.869	0.828	0.835	0.912
	GLB	0.832	0.872	0.829	0.836	0.916
Time 4	Omega	0.862	0.868	0.846	0.843	0.916
	Alpha	0.861	0.866	0.845	0.843	0.914
	GLB	0.862	0.868	0.846	0.843	0.916

F1 = Student–Student Relations; F2 = Student–Staff Relations; F3 = Academic Emphasis; F4 = Shared Values and Approaches; F5 = School Identification.

## Data Availability

The dataset for the study is available from the corresponding author upon reasonable request due to ethical restrictions.

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
