# Peer review of "Longitudinal Measurement Invariance of the Dual School Climate and School Identification Scale (SCASIM-St15) in Chilean Adolescents"

_behavsci, 2025, doi:10.3390/bs15060750_

Round 1
Reviewer 1 Report
Comments and Suggestions for Authors
An assessment of the longitudinal invariance of a school climate measures is an important addition to the body of work. I had the following suggestions
- on page 4 when discussing work on. school climate and well-being please examine Klik et al (2023) in addition to Yin et al. (2024).
- on page 5 last line & should be 'and'
- 3. page 7 can a reference be added for the "thresholds" and established criteria for the changes in RMSEA, CFI and TLI.
- F1, F2, F3 and F4 seemed to have higher reliability coefficient sat Time 4 is there nay explanation for this?
- On page 9 there is an additional paper on school climate/school identification and bullying by I Turner et al., 2014 School Psychology Quarterly.
Reviewer 2 Report
Comments and Suggestions for Authors
Dear Author(s)
your paper is a very good piece of work, well-written and well-structured. It can be improved in the following areas:
- Abstract and Participants: Clarify the number of participants in your study across all satges/waves.
- Introduction: (p. 2, 2nd paragraph) In your argument that negative school climate is associated with high risk of discrimination against LGBTQ+ adolescents, I suggest you add other vulnerable adolecents (i.e. adolescents with special educational needs, or of immigrant backgrounds, overweight adolescents, adolescents from single-parent families, adolescents with health issues, etc). Additionaly, you much refer to loneliness, victimization and bullying and their link to school climate. (p. 2, 4th paragraph) This paragraph must be moved earlier to the Introduction, since it deals with the definition of school climate. (p. 3, 1st paragraph) What is missing here is a solid theoretical base for the connection between the concepts of school identity and school climate. (p. 4, 3rd paragraph) Again you must incoprorate your arguments about school climate and its impact on adolescence earlier in the Introduction where you first started talking about it.
- At the end of the Introduction, add your Research Hypotheses
- Discussion: Add limitations and detailed implications for school-based intervention.
Reviewer 3 Report
Comments and Suggestions for Authors
This is an interesting article on school climate. Micro and macro climate is very important for any educational institution as it can be a source of inspiration for education but can also become a barrier to creative and effective work.
The introductory part of the article is well done, the results clearly presented and statistically processed.
I have reservations about Chapter 4 - Discussion.
The discussion of the results is too general and at times it is not clear whether the conclusions or statements are from the research itself or are taken from it. I recommend that Chapter 4 - Discussion be revised to include specific results of similar research and to compare them with the results obtained in the own research.
I also recommend expanding the conclusion of the article to include suggested solutions for creating a good school climate.
The English is understandable, but I will leave the assessment to the experts.
Round 2
Reviewer 3 Report
Comments and Suggestions for Authors
The article has been sufficiently edited and I have no comments.